# GroupBERT: Enhanced Transformer Architecture with Efficient Grouped Structures

## Abstract

Attention based language models have become a critical component in state-of-the-art natural language processing systems. However, these models have significant computational requirements, due to long training times, dense operations and large parameter count. In this work we demonstrate a set of modifications to the structure of a Transformer layer, producing a more efficient architecture. First, we rely on grouped transformations to reduce the computational cost of dense feed-forward layers, while preserving the expressivity of the model . Secondly, we add a grouped convolution module to complement the self-attention module, decoupling the learning of local and global interactions. We apply the resulting architecture to language representation learning and demonstrate its superior performance compared to BERT models of different scales. We further highlight its improved efficiency, both in terms of floating-point operations (FLOPs) and time-to-train.

## 1 Introduction

Deep neural networks have emerged as the leading solution to enabling end-to-end language processing (Hochreiter & Schmidhuber, 1997; Sutskever et al., 2014; Chung et al., 2014). Recently, the Transformer model based on the self-attention mechanism (Vaswani et al., 2017) has become the most promising architecture for language applications, especially when used as a pre-trained foundation model (Devlin et al., 2018; Radford et al., 2019; Brown et al., 2020; Bommasani et al., 2021). Attention based models are also increasingly showing promising results for established applications in domains different from natural language processing (Dosovitskiy et al., 2020).

Complementary to the Transformer's improved ability to model long-range dependencies in sequences is its superior potential to scale to larger sizes (Kaplan et al., 2020) and its suitability for execution on existing accelerators. This makes these models favoured over traditional recurrent language models. Given the increased computational demand of these models, there is a growing and pressing interest to develop more efficient architectures (Strubell et al., 2019). Some previous proposals were able to reduce the computational burden of the Transformer with improved task performance, but often with a corresponding slower execution, as will be discussed further in Section 4.

We demonstrate a set of modifications to the the structure of the Transformer layer that improve FLOP utilization by the encoder stack. The proposed *GroupBERT* model relies on grouped matrix multiplications and convolutions, and delivers a more efficient version of BERT, superior in both task performance and computation. GroupBERT utilizes grouped operations in a novel way which makes the model more FLOP efficient. However grouped operations are characterised by a reduced computational load for a given memory access (Masters et al., 2021), i.e *arithmetic intensity*. This property would make them undesirable for traditional accelerators, which rely on large dense computations and reduced memory access. For this investigation we use the IPU (Jia et al., 2019) hardware, since its architecture uses on-chip SRAM for model execution, making it suitable to explore the potential of these efficient building blocks.

We achieve a further task performance boost by increasing the depth of the model as GroupBERT extends each Transformer layer to contain four modules: one multi-head attention (MHA), one grouped convolution module, and two grouped feed-forward modules (GFFN). The MHA and grouped convolution modules process token information along the sequence dimension, and each is followed by the general computation GFFN module. While there are twice as many modules in

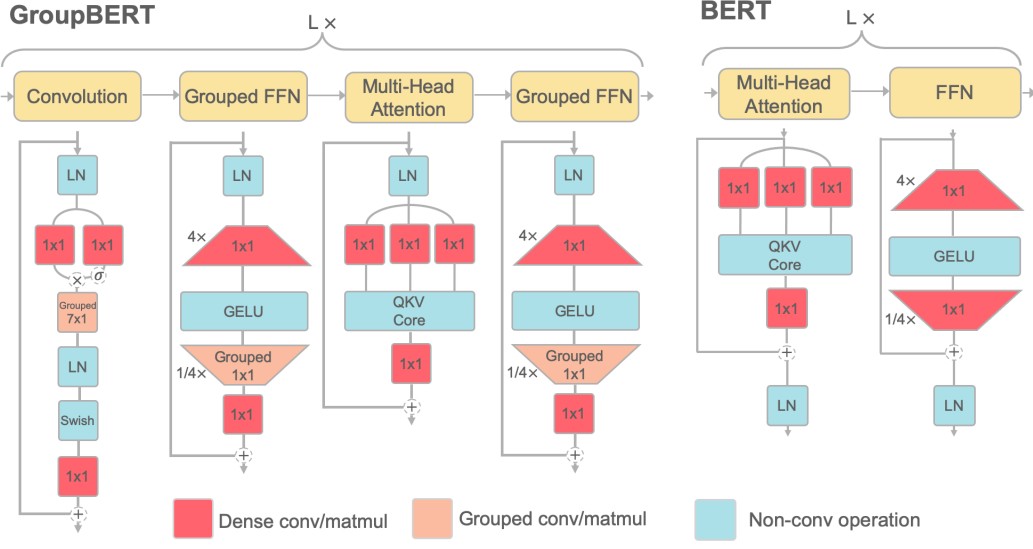

Figure 1: Schematic representations outlining the difference between GroupBERT and BERT structures. A single GroupBERT layer has double the number of modules.

the proposed GroupBERT layer, the overall increase in computation is modest as we utilize sparse grouped operations, for a total FLOP increase of about $60\%$.

Not only does GroupBERT deliver better performance per FLOP, but it is also executed faster as measured in total time-to-train. By employing both attention and convolution, the model has components dedicated to both short and long-range interactions, making a more efficient use of the more expensive attention mechanism. We also utilize the parameters of GroupBERT more efficiently during training, by discarding dropout for pre-training on a large corpus of text and by improving stability to use higher learning rates. With all these innovations, GroupBERT Base is only slightly larger than BERT Base, yet it achieves better validation MLM loss than BERT Large using less than half of its FLOPs.

## 2 ARCHITECTURE

In this work, we propose an efficient modification of the Transformer encoder layer called GroupBERT. The original Transformer layer consists of two modules: multi-head attention (MHA) and feed-forward network (FFN). Each of these modules also includes dropout, a shortcut connection, and layer normalization (Srivastava et al., 2014; He et al., 2016; Ba et al., 2016). GroupBERT includes four modules in every layer, as illustrated in Figure 1. We add a convolution module in sequence with the MHA to efficiently model local interactions between tokens and to allow the attention mechanism to focus on long-range interactions. We then complement every sequence processing block with a dedicated fully-connected module. For better efficiency, we introduce grouped projections to the FLOPs intensive FFN module, making the layer structure more FLOP efficient.

### 2.1 GROUPED FEED-FORWARD MODULES

The FFN module plays a crucial part in the unparalleled task performance of Transformers (Dong et al., 2021; Lee-Thorp et al., 2021). Although it is an essential complement to sequence processing modules it introduces a significant computational burden, as two thirds of the FLOPs are concentrated in the FFN module. To make this integral part of the transformer more lightweight we utilize structured sparsity in a form of sparsely grouped matrix multiplication. Consider a dense matrix multiplication of matrices $\mathbf{H} \in \mathbb{R}^{a \times b}$ and $\mathbf{W} \in \mathbb{R}^{b \times c}$:

$$(\mathbf{HW})_{i,j} \triangleq \sum_{n=1}^{b} h_{i,n} \cdot w_{n,j} \tag{1}$$

A sparsely grouped version of $\mathbf{W}$ corresponds to a block diagonal matrix $\mathbf{W}^{(G)}$ with $G$ groups, a matrix of similar dimension to $\mathbf{W}$ and a sparsity ratio of $1/G$. An equivalent alternative formulation of a block-diagonal matrix is a grouped convolution for a 1-dimensional $1 \times 1$ convolution (Iandola et al., 2020). This reduces the number of stored parameters, and can be implemented efficiently without zero-multiplication as:

$$(\mathbf{HW}^{(G)})_{i,j} \triangleq \sum_{n=1}^{b} h_{i,n} \cdot w_{n,j} = \sum_{n=1}^{b/G} h_{\,i,\,n\,+\,\frac{b}{G}\,\cdot\,\lfloor j-1\,/\,\frac{c}{G}\rfloor} \cdot w_{\,n\,+\,\frac{b}{G}\,\cdot\,\lfloor j-1\,/\,\frac{c}{G}\rfloor,\,j} \tag{2}$$

We propose a novel scheme to utilize grouped transformations in a transformer FFN module. Our first finding is that parameters in the expanding FFN matrix contribute more to task performance, and sparsity is particularly damaging for these fan-out matrices. The second matrix is less sensitive to parameter reduction due to the sparse input and the reduction of projection dimension. Therefore, introducing sparsity in the second matrix results in a Pareto efficient balance between compute and task-performance. Our second finding is that the locality constraint of grouped projections on the hidden dimension is detrimental to the model. However we remove this constraint by using an output linear projection. This is similar to the output projection matrix used in the MHA block, where each attention head acts only on a slice of the hidden dimension. We find the optimal value for the number of groups to be $G = 4$, bringing the parameter count of GFFN to be 75% of its dense counterpart, while also delivering a more Pareto efficient task performance (Fig 6). Alternative schemes of applying grouped transformations (Iandola et al., 2020; Mehta et al., 2021; Jiang et al., 2020) (Figure 2) in a transformer fail to outperform the baseline in a setting of BERT pre-training (see Section 3.3).

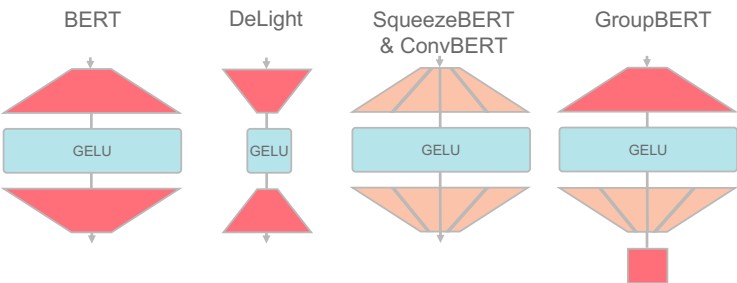

Figure 2: Schematic diagram showing the comparison of FFN implementations of different transformer variants that rely on grouped matrix multiplications.

## 2.2 CONVOLUTION BLOCK

Sequential locality plays an important role for contextualizing tokens in language models. At the same time, long-range interactions have proven to be vital for state-of-the-art performance. Transformers inherently support long-range content-based interactions via self-attention and usually incorporate a form of positional encoding, allowing attention to also capture position-based interactions (Dai et al., 2019). Although this gives self-attention strong representational power, a convolution is a more efficient implementation of strictly local, position-based fusion. For this reason we adopt a dedicated convolutional module to improve overall efficiency.

The design of our convolution module is similar to Gulati et al. (2020), in which convolutions were introduced into a speech recognition Transformer. We apply a gate consisting of a pointwise convolution followed by a Gated Linear Unit (GLU) that has been beneficial in language applications (Dauphin et al., 2017; Wu et al., 2019a; 2020). Unlike Gulati et al. (2020), we use grouped convolutions in place of depthwise convolutions to add representational capacity. We find that the best trade-off

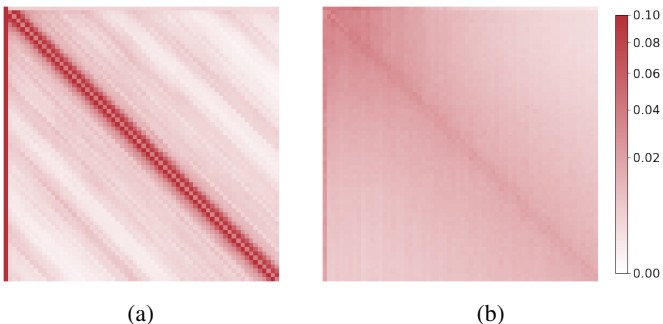

(a)                                                    (b)

Figure 3: Attention maps of a single head from the first MHA module of BERT Base (a) and Base + Conv (b) (Section 3.5), averaged over $10^3$ validation sequences to show the sequential locality preference of a typical attention head (see Appendix C for all heads). Each row corresponds to a query token position, each column to a key position. Adding the convolution module encourages attention layers to learn long-range content-based interactions over short-range position-based interactions.

between task performance and computational cost is achieved by using a grouped convolution with group size 16 and kernel size 7, computed over the sequence dimension. The module also includes an additional layer normalization and a Swish activation (Ramachandran et al., 2017).

With this module included, fewer attention heads show a strong locality preference since such interactions are readily captured by convolutions. This effect is visible in the attention maps of Figure 3, showing weaker locality in the model that includes convolutions. To measure this effect quantitatively, we calculate the entropy across target positions for each head and source position. We then average, and normalize by the maximum possible value (see Appendix C). For this measure, zero means that every head attends to a single position exclusively, while one means that every head is position agnostic, although there could still be a joint position and content term. BERT Base has an average entropy ratio of $0.75$ and BERT Base + Conv has $0.92$, indicating a shift of positional fusion work from attention to convolution.

### 2.3 EFFICIENT PARAMETER UTILIZATION

In line with earlier research on the Transformer architecture (Wang et al., 2019; Liu et al., 2020; Xiong et al., 2020), we move layer normalization (Ba et al., 2016) from its position after the module's residual (*"postnorm"*) to the first position within each residual block (*"prenorm"*). While this modification does not directly improve task performance, it stabilizes training and allows the use of a larger learning rate that would otherwise trigger the model with postnorm to diverge. We increase the learning rate by a factor of $4\times$ compared to the postnorm baseline.

Similarly to Lan et al. (2020), we find the use of dropout to be detrimental to the pre-training stage. Due to the substantial size of the dataset, this kind of regularization is not required. While removing dropout yields improvements to the pre-training loss, this does not apply to downstream tasks that rely on smaller datasets. Consequently, we include dropout only when fine-tuning on supervised tasks, that have smaller datasets than the pre-training corpus.

## 3 RESULTS

To evaluate the architecture modifications, we chose BERT (Devlin et al., 2018) pre-training and fine-tuning. The large dataset and challenging training objective mean that task performance improves consistently with model size (Lan et al., 2020) and the risk of over-fitting is reduced. This makes it possible to clearly distinguish architecture modifications that benefit efficiency.

Our evaluation of GroupBERT for language representation learning shows that the architecture is:

1. Training FLOP-efficient across a range of model sizes (Sections 3.3, 3.4).
2. Training time-efficient across a range of compute budgets (Sections 3.3, 3.4).

    3. Improved by each constituent part (Section 3.5).

## 3.1 EXPERIMENTS

Each experiment consists of two pre-training phases and a fine-tuning phase consisting of multiple training runs, started from the pre-trained model. All phases use the AdamW optimiser (Loshchilov & Hutter, 2019), with $\beta_1 = 0.9$, $\beta_2 = 0.999$, $\epsilon = 10^{-6}$. The learning rate follows a linear warm-up decay schedule, whereby the warmup phase lasts for $\min(10^4, 0.1 \cdot$ total steps$)$ steps, and the peak learning rate depends on the training phase and model size. The model is defined over a vocabulary of $30522$ WordPiece tokens (Wu et al., 2016). Weights are initialized using a truncated normal distribution of standard deviation $0.02$. For all experiments we use 2 Graphcore M2000 IPU systems.

*Pre-training phase one* optimises the Masked Language Model (MLM) and Next-Sentence Prediction (NSP) loss for corrupted sentence pairs. Masked and padded sequences of length $128$ are grouped into batches of approximately $512$ sequences, with slight variations depending on the model size (see Appendix A). The model is trained for 10 epochs of Wikipedia + BookCorpus (Zhu et al., 2015), corresponding to approximately $8 \cdot 10^5$ optimisation steps. For all experiments with GroupBERT, baseline BERT and all other models tested, the learning rate is set to the value that produces the best validation loss. *Pre-training phase two* uses sequence length $384$, 5 epochs, and approximately $2 \cdot 10^5$ optimisation steps.

*SQuAD 1.1 fine-tuning* (Rajpurkar et al., 2016) adds a token span prediction layer and the whole model is fine-tuned to perform extractive question answering. Training uses target batch size 32 and we train for 2-3 epochs with various learning rates (Appendix B) and report results for the best hyperparameters setting. We report F1 and Exact match scores, which show higher variance than MLM loss values. On the grounds of larger variance, we fine-tune each pre-training checkpoint five times using different seeds for every hyperparameter setting. Fine-tuning has been shown to be quite a brittle process in recent studies (Dodge et al., 2020; Zhang et al., 2021; Mosbach et al., 2021). In particular, many instabilities are caused by fine-tuning without using bias correction, an implementation that was adopted following the original experimental setup of BERT. This omission in the optimizer was observed to cause a collapse of the training process. For this reason, we included a bias-correction term to the AdamW implementation for fine-tuning.

## 3.2 IMPLEMENTATION

In this work we leverage Graphcore's Intelligence Processing Unit (IPU) (Jia et al., 2019), as it allows to examine a wider variety of techniques, centered around improving computational efficiency. In particular the IPU can execute grouped operation efficiently. We train all models on Graphcore Mk2 IPU clusters using a combination of pipeline model parallelism and data parallelism (Harlap et al., 2018). To make efficient use of compute resources, we increase cluster size and pipeline depth based on model size. We maximise the "compute" batch size executed by each node to achieve maximum throughput, and choose the pipeline gradient accumulation count to achieve a target "global" batch size (see Appendix A for more details).

All parameters, activations, and optimiser states are stored and processed in IEEE 754 half-precision floating-point. This excludes the Adam weight update calculation and variance state, the loss calculation from logits and temporary variables that require higher precision. Higher precision temporary variables are used by layer normalization, softmax, and the accumulation of partial matrix multiplications. Our models are implemented in TensorFlow and are publicly available[1].

## 3.3 PRE-TRAINING

BERT type models are notoriously difficult to evaluate as most of the computation is devoted to performing self-supervised pre-training on unlabeled corpora of text. As this process is the most costly one, we are primarily interested in evaluating solely this part of the model training separately. Taking example from other studies (Kaplan et al., 2020; Tay et al., 2021), we choose MLM loss on the validation dataset as the most salient criterion for task-performance as it captures language in its generality.

---

[1]Github: https://github.com/neurips_authors/example

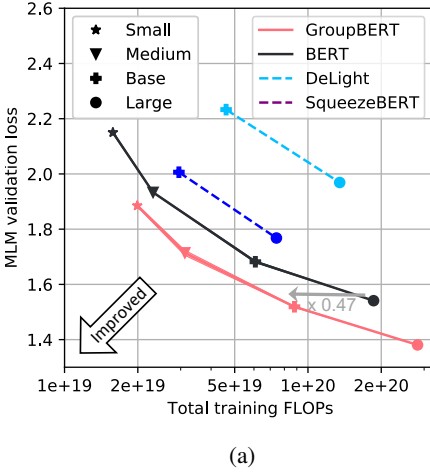 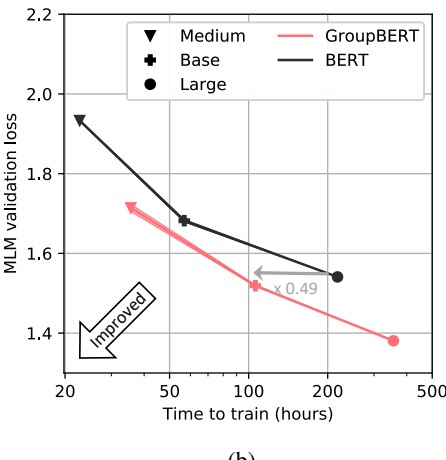

(a)                                                          (b)

Figure 4: The GroupBERT model architecture is Pareto efficient on pre-training masked language modeling task. As well as comparing against the original BERT baseline, we tested transformers that use grouped transformations like SqueezeBERT and the uniform version of DeLight. For all models we tuned the learning rate to maximize their task performance. (a) Mean MLM validation loss ± standard deviation plotted against the total pre-training FLOPs. (b) Mean MLM validation loss ± standard deviation plotted against the total time for pre-training.

In Figure 4 we report the MLM loss measured on the validation dataset for GroupBERT and a range of other models of different sizes, including baseline BERT, DeLighT and SqueezeBERT. Evaluating task performance in the against FLOPs (Figure 4a) presents a theoretical picture, which is completely agnostic to implementation or type of hardware used. To complement the theoretical viewpoint of performance weighted against FLOPS, we also assess speed of execution, highlighting that GroupBERT additionally saves overall time-to-train (Figure 4b). Figure 4 summarizes the practical benefits of using GroupBERT in terms of both FLOPs and training time to reach a desired task performance. By plotting many model sizes of the same family, the efficiency of GroupBERT is clearly visible. The GroupBERT Base model outperforms baseline BERT Large, but uses only 47% of its resources and trains in half the time.

## 3.4 FINE-TUNING

Both the theoretical and the practical Pareto improvements achieved by GroupBERT on MLM evaluation loss translate to the SQuAD fine-tuning task (Figures 5 and Appendix B). Not considering

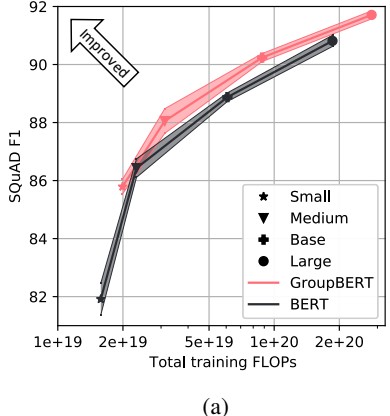 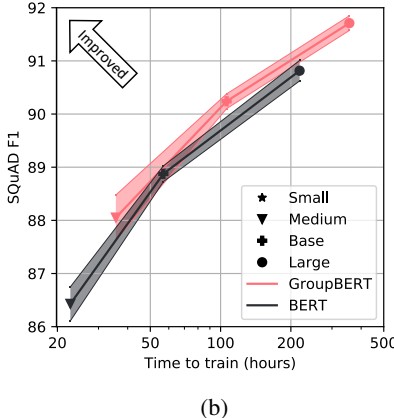

(a)                                                          (b)

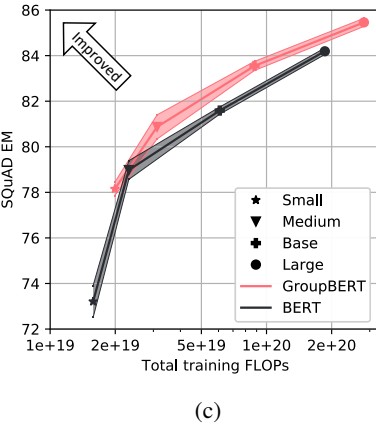 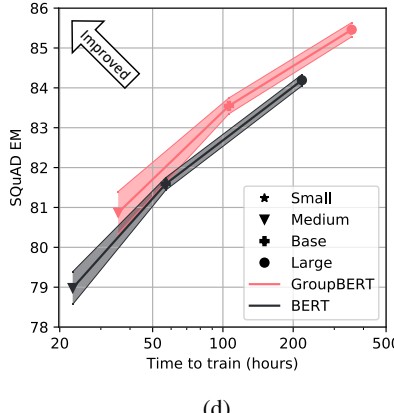

(c)                                            (d)

Figure 5: Median SQuAD F1 and Exact Match scores ± standard deviation of the GroupBERT and BERT model families plotted against the total pre-training FLOPs (a,c) and against the total time to pre-train (b,d).

the additional dropout in fine-tuning, the cost for training on downstream tasks for different model sizes is proportional to the cost for pre-training, while a few orders of magnitude smaller. Hence, FLOPs and time-to-train as measures of training cost are given for pre-training, as fine-tuning takes a comparatively negligible amount of resources.

## 3.5 ABLATION STUDY

To verify the efficiency of the different model changes, we study their individual contribution to the validation MLM loss improvement as well as their cost in terms of parameters and FLOPs (Table 1).

Some of the ablation modifications that we make add extra parameters that trigger a divergence during model training when using the best learning rate. Therefore for the ablation study we use the BERT Base model with the next-best learning rate as the main comparison point, to allow the use of the same learning rate throughout.

We measure the performance of each modification with the Pareto improvement of the corresponding model, defined here as the difference in MLM loss with respect to the baseline models for the same FLOPs count. This is visually represented in Figure 6 by the vertical distance between the ablation point and the baseline interpolation line.

Table 1: Ablation study of the contribution of different model modifications.

| MODEL | PARAMETERS | TRAINING FLOPs | LR | MLM LOSS | IMPROVEMENT |
|---|---|---|---|---|---|
| BERT Base | 110.1M | 6.1e19 | 2e-4 | 1.679 | 0.018 |
| BERT Base | 110.1M | 6.1e19 | 1e-4 | 1.697 | - - - |
| Prenorm | 110.1M | 6.1e19 | 8e-4 | 1.677 | 0.020 |
| No Dropout | 110.1M | 6.1e19 | 1e-4 | 1.651 | 0.046 |
| Convolution | 132.4M | 7.3e19 | 1e-4 | 1.613 | 0.058 |
| 2 GFFNs | 138.5M | 7.6e19 | 1e-4 | 1.660 | 0.005 |
| 2 GFFNs + Conv | 160.8M | 8.8e19 | 1e-4 | 1.573 | 0.072 |
| GroupBERT Base | 160.8M | 8.8e19 | 8e-4 | 1.516 | 0.126 |

By carrying out individual additive ablation, we are able to see the effect of every model modification in isolation. Furthermore, we investigate the interaction of adding both the convolution module and two GFFN modules. We observe that every module that mixes token information gets significantly more efficient when used with a matching GFFN module. While using two GFFN modules increases

the performance in line with the baseline model, using it in conjunction with convolution increases its Pareto improvement.

Figure 6: Ablation results showing the contribution of individual model modifications. Points where we changed the method of training have the same number of FLOPs as the original BERT Base model. Other improvement add some extra FLOPs. Ablation experiments were run with three different random seeds and all points include a standard deviation.

## 4 RELATED WORK

In language modelling, modern attention based architectures utilize most of their non-embedding parameters in dense operations, thus driving up the computational requirements of these models. Some studies have already considered the use of grouped operations in Transformer-XL (Dai et al., 2019): DeLighT (Mehta et al., 2021) and DeFINE (Mehta et al., 2020) use cascades of grouped transformations in the encoder and the embedding, respectively, to lower the computational load of the Transformer. While sophisticated in their design, these models fail to generalize in the setting of self-supervised learning on massive amounts of data and do not provide a compelling acceleration for the model execution. SqueezeBERT (Iandola et al., 2020) replaces almost all dense multiplications with grouped transformations in a Transformer layer. This approach shows a significant speedup, but loses too much task performance if pre-trained without distillation from a dense parent model (Hinton et al., 2015). In this work, we present a more balanced implementation of grouped transformations, delivering both performance and speed.

Grouped transformations have been prevalent in Convolutional Neural Networks (CNN), initially used for their parameter efficiency, starting with Alexnet (Krizhevsky et al., 2012). They have been further popularized by the ResNeXt architecture (Xie et al., 2017). More recent CNN architectures implement depthwise separable convolutions, which are a special case of grouped convolutions with group size equal to one (Howard et al., 2017; Tan & Le, 2019). In addition to being beneficial for resource utilization, the use of grouped transformation was found to be particularly well suited to convolutional neural network architectures (Ioannou et al., 2017).

The Transformer model was initially introduced as a parallelizable solution for supervised sequence-to-sequence learning in the field of natural language processing. These models have become ubiquitous with the use of self-supervised approaches (Devlin et al., 2018; Radford et al., 2019). While attention based models have superior performance due to their ability to model long-range interactions, this characteristic is also what makes the attention mechanism costly, especially when used on long

sequences. Many studies have successfully managed to make the attention mechanism more efficient (Tay et al., 2020; Fournier et al., 2021). However, its general use across the model can be redundant, as some attention heads in practice reduce to convolutions to model local interactions (Cordonnier et al., 2020) and duplicate other attention heads, making them redundant (Michel et al., 2019). Using convolutions directly inside a Transformer has been shown to make attention more focused on long-range interactions, thus enabling more of its capacity to be used for that specific purpose (Wu et al., 2019b; Jiang et al., 2020; Wu et al., 2020). While the incorporation of convolutions in Transformers was previously investigated only for smaller models with no more than 100M parameters, we look at scaling to a much wider range of model sizes, from 30M to 500M parameters.

## 5 DISCUSSION

The improved practical efficiency of GroupBERT reduces the cost of model training. This provides access to better performing language models for a broader range of users, both in the research community and for industrial applications. Moreover, the advantage of GroupBERT over the BERT baseline model family increases with model size, and therefore reduces the cost for training large and highly accurate language models. This observed scaling behaviour indicates the capability of GroupBERT to maintain a material advantage for even larger language models. This can be a valuable improvement for state-of-the-art research, relying on models that are increasing in size exponentially. With the latest breakthroughs using significant amounts of energy and hardware resources (Brown et al., 2020), efficiency improvements can translate into significant savings.

The BERT models represent not only a step forward for the NLP research community, but also a tool heavily exploited in day-to-day operations of large corporate entities. Hence, improving the efficiency of these models that are used in production environments can potentially result in significant power savings worldwide, thus limiting greenhouse gas release into the atmosphere. Due to the same factor, all experiments that were carried out for this investigation were done using $100\%$ renewable electricity, making this work carbon neutral.

In this paper we propose a general technique for improving self-supervised Transformers. However, we only investigate its effectiveness on the English language, due the ease of comparison with other studies using the same datasets. A potential ethical concern is the increasing dominance of language models trained on the English language, which may be detrimental to the efforts of preservation of endangered languages. Another limitation of this study is the use of the BookCorpus dataset as part of the pre-training process. Pre-training models on this dataset was a standard practice during the course of the investigation, but since then was found to be of questionable ethical value (Bandy & Vincent, 2021).

We have relied on structured weight sparsity to increase the efficiency of BERT and have created more diversity of building blocks within the self-supervised encoder by complementing the attention block with a convolution module. Our aim is to further build upon these ideas, and construct models that are even more general in their application to different domains and that can be efficiently executed. To do so, we see potential to rely on other forms of sparsity, including dynamic weight sparsity (Evci et al., 2019) and conditional activation sparsity, improving the capability to handle multiple languages and data domains within the same architecture (Fedus et al., 2021).

## 6 CONCLUSION

In this study, we present GroupBERT: an enhanced Transformer architecture that is shown to be a more efficient alternative, with up to $2.1\times$ efficiency gain in terms of both FLOPs and time-to-train. We achieve these improvements by adding a dedicated grouped convolution module to every layer and using grouped transformations to reduce the density of fully connected layers. The proposed model achieves better results on both pre-training and fine-tuning tasks, and applies to a wide range of model scales.

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

## A    EXECUTION SCHEME

In pipeline parallelism, a model is divided between multiple accelerators. Pipeline parallelism is a form of model parallelism where execution is decomposed into a sequence of stages. At a given instant, every stage is processing a different and independent partition of the model, before passing it on to the next stage. For the majority of our experiments we split the model between four chips, three for encoder layers and one for embedding, projection and loss. However GroupBERT Large has a larger memory footprint, making an eight-chip pipeline the most efficient for execution. To create computational parity between all models, we replicate four-chip pipelines twice for data parallel training, resulting in eight chips being used for all experiments.

In the context of pipelined training, we distinguish between compute batch size and global batch size. The general formula for global batch size is:

Global batch size = Replicas × Accumulation factor × Pipeline depth × Compute batch size

Here the compute batch size is the largest portion of the global batch at every pipeline stage, which is always maximized for efficient resource utilization. The pipeline depth is given by the total number of stages in the forward and backward passes. Therefore, the accumulation factor is the only variable independent from the model structure and pipeline layout, which we can change to get a desired global batch size. We target the use of a global batch size $500 \pm 20$ for all experiments. This value is not exact and has to vary slightly for different model sizes, as the pipeline layout dictates the exact global batch size that can be used. However, we have not observed any significant effect of these variations on task-performance.

Table 2: Overview of the different model sizes used in this study, with the number of parameters used for the BERT baseline model and our GroupBERT architecture.

| MODEL SIZE | LAYERS | HIDDEN SIZE | PARAMS BERT | PARAMS GROUPBERT |
|---|---|---|---|---|
| Small | 4 | 512 | 29.1M | 37.6M |
| Medium | 8 | 512 | 41.7M | 56.9M |
| Base | 12 | 768 | 110.1M | 160.8M |
| Large | 24 | 1024 | 336.2M | 515.5M |

Table 3: Pre-training hyperparameters for both BERT and GroupBERT model families.

| FAMILY | HYPERPARAMETER | SMALL | MEDIUM | BASE | LARGE |
|---|---|---|---|---|---|
| BERT | Learning rate | 4e-4 | 2e-4 | 2e-4 | 1e-4 |
| | Batch size | 512 | 480 | 480 | 512 |
| GroupBERT | Learning rate | 3e-3 | 1.5e-3 | 8e-4 | 4e-4 |
| | Batch size | 512 | 480 | 480 | 480 |

## B  FINE-TUNING RESULTS AND HYPERPARAMETERS

Table 4 contains the SQuAD v1.1 fine-tuning results produced from the validation dataset. When fine-tuning GroupBERT models, we always used batch size $32 \pm 2$ (for the reasons outlined in Appendix A), trained for either 2 or 3 epochs, with learning rates being one of the following: {1e-4, 1. 5e-4, 2e-4, 3e-4, 4e-4}. Each model required a sweep to identify the best candidate. The sweep for the baseline BERT models was performed according to range of hyperparamters specified in Devlin et al. (2018).

Table 4: SQuAD v1.1 results, F1/Exact match %.

| SCORE | SMALL | MEDIUM | BASE | LARGE |
|---|---|---|---|---|
| BERT | 73.2 / 81.9 | 79.0 / 86.4 | 81.6 / 88.9 | 84.2 / 90.8 |
| GroupBERT | 76.8 / 84.7 | 80.9 / 88.1 | 83.5 / 90.2 | 85.5 / 91.7 |

## C  ATTENTION MAPS

To visualise the effect of introducing a convolution block into the model, we study the attention maps of each head. An attention map shows the softmax weight between every pair of positions in the sequence. In order to remove the effect of content and focus on position, we average the attention map over $10^3$ validation sequences. We show these maps in Figures 7, 8 and 9, generated after pre-training phase one.

To quantify the locality of an attention head, we define the normalized positional entropy:

$$H(a) = \frac{1}{L \log L} \sum_{i=1}^{L} \sum_{j=1}^{L} a_{ij} \log a_{ij}, \tag{3}$$

where $a_{ij}$ is the attention weight between source position $i$ and destination $j$ and $L$ is the sequence length. We average the normalized positional entropy over all layers and heads to get a single metric describing a trained model. This measures to what extent attention maps are spread out over different positions. Results for Base-sized models are given in Table 5. The models were trained for the ablation study of Section 3.5, showing the effect of individual components on entropy.

Table 5: Average normalized positional entropy for the models of Section 3.5. Models containing convolutions, Conv and GroupBERT, show higher entropy.

| MODEL | NORMALIZED ENTROPY |
|---|---|
| BERT Base | 0.75 |
| Conv | 0.92 |
| GroupBERT Base | 0.89 |

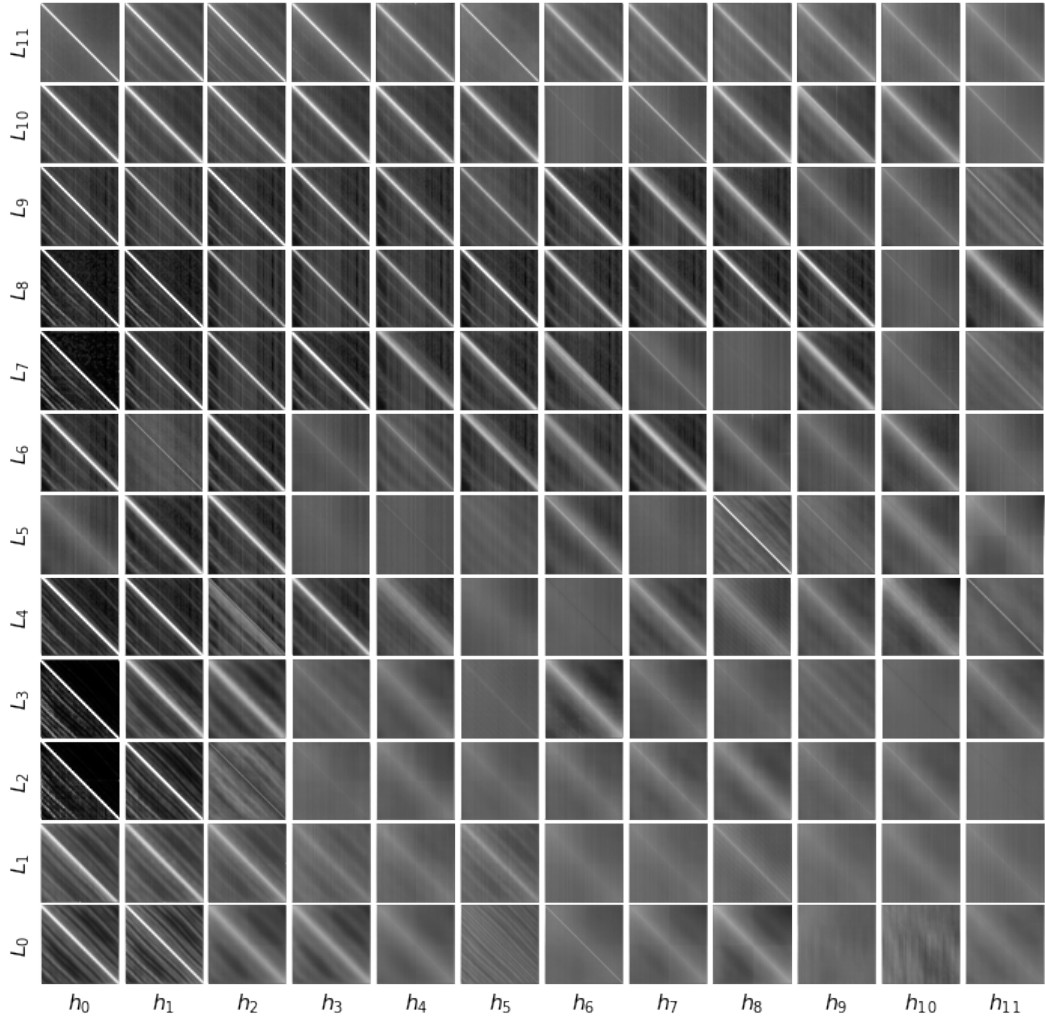

Figure 7: BERT Base attention maps, for all layers $L_0-L_{11}$. Heads $h_0-h_{11}$ in each layer are ordered by positional entropy, Equation 3. Drawn with a value range of $[0, 0.1]$ and gamma of $1/3$.

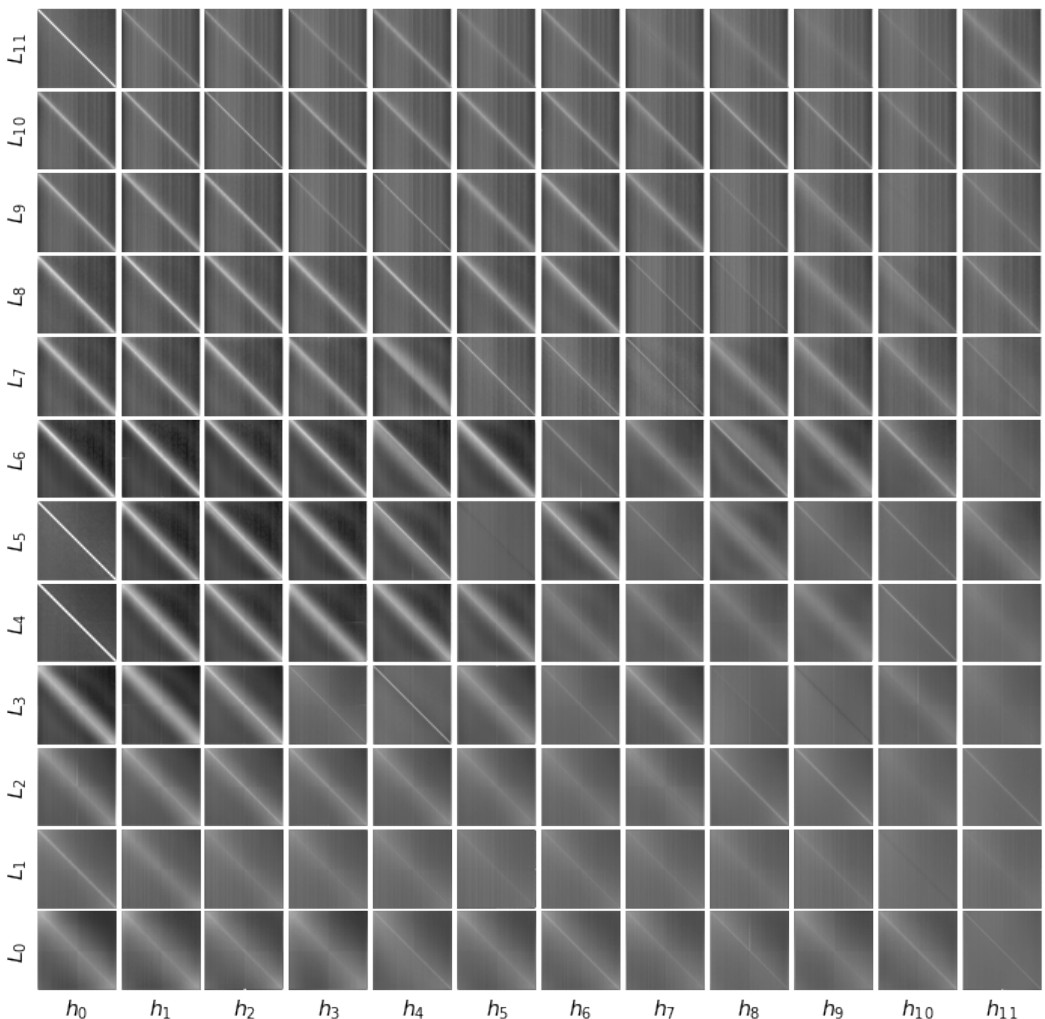

Figure 8: BERT Base + Conv attention maps, generated and drawn as per Figure 7.

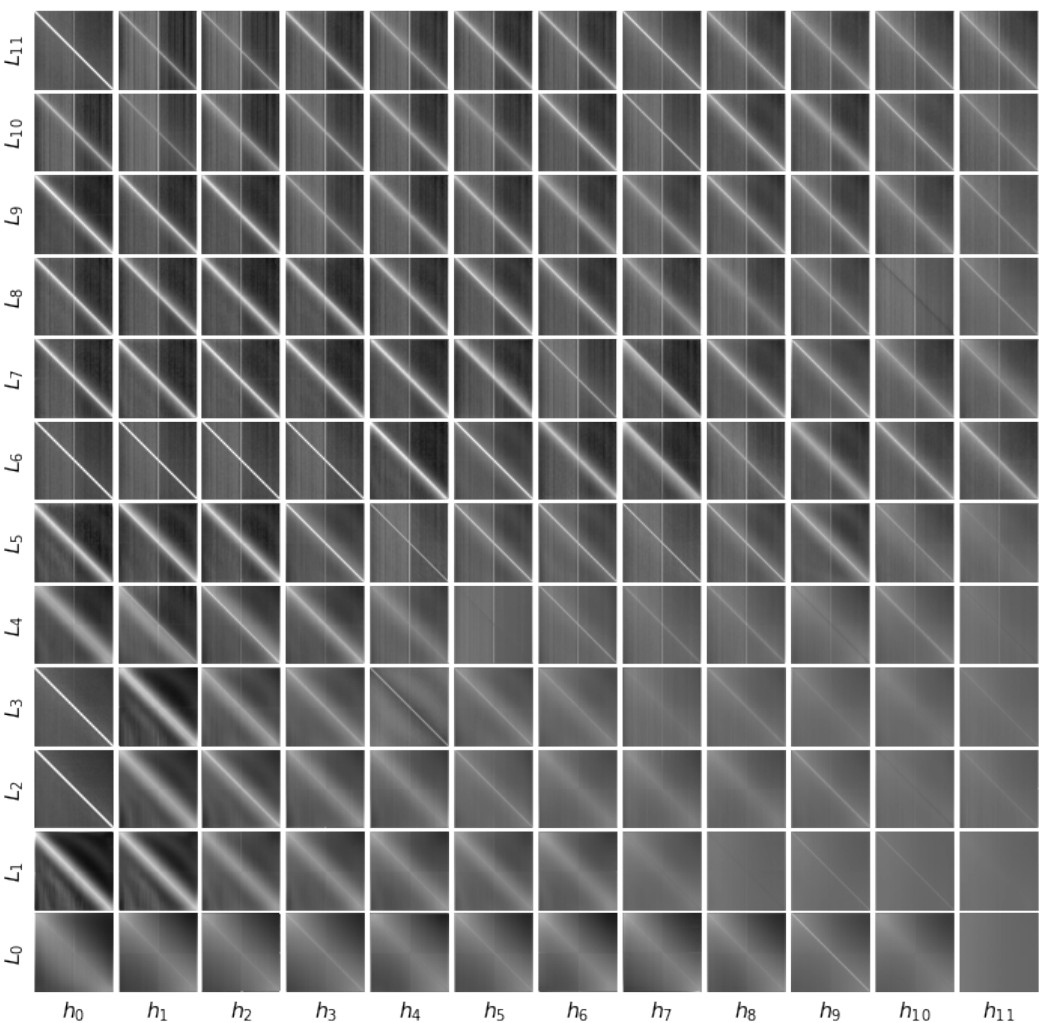

Figure 9: GroupBERT Base attention maps, generated and drawn as per Figure 7.

