# OpenReview forum: "GroupBERT: Enhanced Transformer Architecture with Efficient Grouped Structures"
_ICLR.cc/2022/Conference — ICLR 2022 Submitted_

### Official Review · Reviewer_EcxR · 2021-11-01

**Correctness:** 3
**Technical Novelty And Significance:** 2
**Empirical Novelty And Significance:** 2
**Recommendation:** 3
**Confidence:** 5

**Main Review:**

Strengths:
1. The writing of this paper is good and easy to follow.
2. This paper introduces an interesting grouped feed-forward network for efficient training.
3. Authors introduce the normalized position entropy to quantify the locality of an attention head is also interesting.

Weaknesses:
1. The novelty of this paper is limited. Apply convolution module into self-attention is not a novel idea, and it has been used in many works. For example, in NLP models, it has ConvBERT [a], and in CV tasks, it has [b][c]. What are the advantages or differences of GroupBERT when compared with these works? I think authors should provide these differences, not just compare original BERT.
2. The comparison between GroupBERT and BERT is unfair. You should scale BERT to the same parameters (or the same computations in FLOPs) as GroupBERT for comparison. Otherwise, it cannot argue the contributions is from the proposed method or additional computation. And additionally, GroupBERT introduces too many hyper-parameters, like pre-norm and large learning rate.

Questions:
1. It seems the proposed architecture is hardware-specific. Is IPU necessary for model training and why not select GPU or CPU for testing? Specifically, since one of the targets in this paper is for efficiency (especially for server or embedded device deployment), it is necessary to measure the computations on CPU.
2. Why not report the results on GLUE or SuperGLUE?
3. In addition to the comparison of MLM loss in the training stage, the results on downstream tasks are also very important, especially for pre-trained model. However, authors only compare BERT in downstream tasks (SQuAD v1.1), which lacks convincing. Some other works, including using CNN or for efficiency training, should be considered, like ConvBERT, SqueezeBERT and DeLight, not only the comparison in MLM loss. Besides, why not report SQuAD 2.0 results?
4. It seems the code link (https://github.com/neurips_authors/example) cannot be opened.
5. According to the ablation study in Table 1. it seems that the Conv contributes more benefits than grouped FFN modules. I am wondering how about the performance of the only transformer plus grouped convolution with the same parameters.

[a] Improving BERT with Span-based Dynamic Convolution
[b] CvT: Introducing Convolutions to Vision Transformers
[c] Co-Scale Conv-Attentional Image Transformers

**Summary Of The Paper:**

Pre-trained language models have become a critical component in NLP applications. However, due to its large-scale pre-training computation, pre-trained models are usually too expensive to train. Therefore, in this paper, authors design GroupBERT, which introduces a set of modifications to the structure of the Transformer network. More specifically, they first apply grouped transformation to reduce the computations cost of dense feed-forward layers, and then use grouped convolution to enhance the learning of local and global interactions. Experimental results demonstrate the effectiveness of the proposed method.

**Summary Of The Review:**

This paper introduces GroupBERT, as an efficient architecture for pre-trained models. It designs grouped transformation and grouped convolution for efficient training. However, the experiments of this paper still lack convincing. Most of the experiments are only analyzed on MLM loss in the pre-training stage. However, the downstream task performance is an important metric to measure the quality of the pre-trained models, and this paper only selects SQuAD v1.1 to report performance, which lacks convincing. And the comparisons on downstream tasks only choose BERT as the baseline.

---

> ### Author Response · Authors · 2021-11-19
> **Reply to 14JM**
>
> We thank the reviewer for providing insightful feedback about our work.
>
> To address some of your comments individually:
>
> - “The novelty of this paper is limited. Apply convolution module into self-attention is not a novel idea, and it has been used in many works. For example, in NLP models, it has ConvBERT [a], and in CV tasks, it has [b][c]. What are the advantages or differences of GroupBERT when compared with these works? I think authors should provide these differences, not just compare original BERT.”
>
> The main contribution to our work is the use of grouped transformation instead of dense operations inside of fully connected modules as well as convolutions. Therefore, we narrowed the score of comparisons to other transformers that focused on applying grouped transformations. Figure 4(a) presents a comparison against DeLight and SqueezeBERT that also rely on grouped transformation, but are not competitive when compared to the dense baseline.
>
> - “The comparison between GroupBERT and BERT is unfair. You should scale BERT to the same parameters (or the same computations in FLOPs) as GroupBERT for comparison.”
>
> To draw a conclusive comparison between two architectures we used an extensive family of models, rather than a single model. Between these points we interpolate (on a logarithmic scale) to see if one architecture is better than the other by means of Pareto fronts. Scaling two models to exactly the same FLOPs is feasible, yet not optimal as this can disturb the balance between the width and depth of the model. Such non-optimal scaling can result in degradation of performance, thus we chose to rely on an interpolation between many points of an architectural family and scale models according to Turc et al. [1].
>
> Making a parameter comparison can be done using GroupBERT Base vs BERT Large models. GroupBERT Base requires half the FLOPs and parameters of BERT Large, yet they reach the same pretraining performance.
>
> - “It seems the proposed architecture is hardware-specific. Is IPU necessary for model training and why not select GPU or CPU for testing? ... it is necessary to measure the computations on CPU.”
>
> Model implementation which will be shared post review does not rely on any IPU specific custom ops and can be executed on any hardware that supports TensorFlow library. The reviewer is correct in implicating that throughput is always dependent of the hardware used. However, we believe that properties of different hardware types should not dictate research on efficient machine learning architectures. To add a hardware agnostic measure we present the evaluation against the FLOP count for training the model.
>
> - “Why not report the results on GLUE or SuperGLUE?”
>
> We agree with the reviewer’s comments, and we will provide a range of GLUE results before the end of the discussion period.
>
> - “In addition to the comparison of MLM loss in the training stage, the results on downstream tasks are also very important, especially for pre-trained model. However, authors only compare BERT in downstream tasks (SQuAD v1.1), which lacks convincing. ... Besides, why not report SQuAD 2.0 results?”
>
> We chose MLM loss to provide the comparisons, as it gives a more general metric for language understanding and evaluates the efficiency of pretraining directly, without additional fine-tuning. This approach eliminates additional complexity and is more valuable since pretraining is the most computationally expensive part of bidirectional transformer training. We have evaluated GroupBERT base on SQuAD 2.0 and share the results below:
>
> | Model | SQuAD 2.0 Exact Match | SQuAD 2.0 F1  | FLOPs |
> | ----------- | ----------- | ----------- | ----------- |
> | BERT Base      | 72.7 | 75.7 | 6.1e19 |
> | GroupBERT Base   | 77.3 | 79.1 |8.8e19 |
> | BERT Large   | 78.7 | 81.9 | 1.9e20 |
>
> These results demonstrate that GroupBERT maintains its Pareto efficiency with the more challenging SQuAD 2.0 task. Exact Match for GroupBERT base almost match BERT Large, while saving 50% of time and computation.
>
> - “It seems the code link (https://github.com/neurips_authors/example) cannot be opened.”
>
> We have only provided a placeholder to preserve the anonymity of the authors. A functioning link will be attached to the text post-review.
>
> - “... it seems that the Conv contributes more benefits than grouped FFN modules. I am wondering how about the performance of the only transformer plus grouped convolution with the same parameters.”
>
> Grouped FFNs complement the conv module and increase its advantage over the baseline as seen in “Improvement” column of Table 1. The experiment that is described by the reviewer is part of the ablation study (Convolution has 0.058 improvement) and does not show as much improvement as a configuration with Grouped FFNs (2 GFFNs + Conv  0.072).
>
> [1] Turc et al., Well-Read Students Learn Better: On the Importance of Pre-training Compact Models, 2019.

---

> > ### Author Response · Authors · 2021-11-22
> > **GLUE results**
> >
> > We have conducted a range of experiments with BERT and GroupBERT families of models on GLUE datasets, which indicated there is a technical issue in our implementation of glue downstream tasks. We are working to fix this and will update the camera ready version with these scores.

---

### Official Review · Reviewer_14JM · 2021-11-02

**Correctness:** 3
**Technical Novelty And Significance:** 2
**Empirical Novelty And Significance:** 2
**Recommendation:** 5
**Confidence:** 4

**Main Review:**

Strength:

The application of grouped FNN is well-motivated as two-thirds of the FLOPs are concentrated in the FFN module. It is natural to use grouped FFN to reduce the computation costs of FFN.

The writing is clear and easy to follow.

The proposed approaches achieve higher results than previous Transformer variants with grouped matrix multiplications.

Weakness:

Since previous approaches have applied grouped matrix multiplications in Transformer, the idea of GROUPBERT is not very novel to me.

There are other efficient Transformer variants, like BERT quantification and tiny BERT with knowledge distillation. They can achieve massive speedups while the proposed approach achieves up to 2x speedups. I am wondering whether the proposed architecture can still boost the performance if applied on these extremely small networks.

The comparison focuses on training/validation losses, rather than downstream results, which is not very convincing to me.  Previous methods usually report results on the GLEU benchmark to verify that their methods can achieve better performance on diverse NLP tasks. This work only reports results on SQUAD. It would be better to add more solid results on various downstream datasets.

**Summary Of The Paper:**

The authors propose a new Transformer architecture. Compared the the traditional Transformer, the proposed architecture has two new features.:1) grouped FFN and 2) a convolution module. The grouped FFN is an efficient replacement of the original FFN. The convolution module is responsible for capturing local dependencies. Experiments demonstrate that the proposed variant achieves up to 2x efficiency gain in terms of both FLOPs and time-to-train.

**Summary Of The Review:**

The idea is well-motivated and easy to follow. It would be better to report more solid comparisons on popular GLUE datasets.

---

> ### Author Response · Authors · 2021-11-19
> **Reply to 14JM**
>
> We thank the reviewer for providing insightful feedback about our work.
>
> To address some of your comments individually:
>
> - “There are other efficient Transformer variants, like BERT quantification and tiny BERT with knowledge distillation. They can achieve massive speedups while the proposed approach achieves up to 2x speedups. I am wondering whether the proposed architecture can still boost the performance if applied on these extremely small networks.”
>
> In this study we specifically investigated undistilled models, to measure the cost-performance trade-offs of pre-training models from scratch. Distillation requires a large pretrained model which can be very expensive and in-efficient to train in the first place. However, similar to models from Turc et al. [1] GroupBERT is a family of models ranging across different sizes and a smaller GroupBERT model suitable for low-power budget can be trained with a distilling parent. However, those experiments were out of scope for this investigation.
>
> - “The comparison focuses on training/validation losses, rather than downstream results, which is not very convincing to me. Previous methods usually report results on the GLEU benchmark to verify that their methods can achieve better performance on diverse NLP tasks. This work only reports results on SQUAD. It would be better to add more solid results on various downstream datasets.”
>
> We agree with the reviewer’s comments, and we will provide a range of GLUE results before the end of the discussion period. In the meantime, we were able to collect SQuAD 2.0 values for GroupBERT Base and compare them with the baseline:
>
> | Model | SQuAD 2.0 Exact Match | SQuAD 2.0 F1  | FLOPs |
> | ----------- | ----------- | ----------- | ----------- |
> | BERT Base      | 72.7 | 75.7 | 6.1e19 |
> | GroupBERT Base   | 77.3 | 79.1 |8.8e19 |
> | BERT Large   | 78.7 | 81.9 | 1.9e20 |
>
> These results demonstrate that GroupBERT maintains its Pareto efficiency with the more challenging SQuAD 2.0 task. Exact Match for GroupBERT base almost match BERT Large, while saving 50% of time and computation.
>
> [1] Turc et al., Well-Read Students Learn Better: On the Importance of Pre-training Compact Models, 2019.

---

> > ### Author Response · Authors · 2021-11-22
> > **GLUE results**
> >
> > We have conducted a range of experiments with BERT and GroupBERT families of models on GLUE datasets, which indicated there is a technical issue in our implementation of glue downstream tasks. We are working to fix this and will update the camera ready version with these scores.

---

### Official Review · Reviewer_8smw · 2021-11-03

**Correctness:** 3
**Technical Novelty And Significance:** 1
**Empirical Novelty And Significance:** 1
**Recommendation:** 3
**Confidence:** 4

**Main Review:**

Strengths:
- Evaluation on total training FLOPs vs. MLM validation loss looks reasonable. GroupedBERT is more efficient, that is agnostic to implementation or the hardware type. However, the training time vs. validation is not intuitive as grouped operation can hurt step time on a different hardware, such as a GPU or a TPU.

Weaknesses:
- The ideals look trivial to the reviewer. Grouped FFN is not new and the structured sparsity formulation is not surprising. The hardware performance implication of the structured sparsity is not clear, as the paper is evaluated on a novel hardware accelerator. What are the performance implications when evaluating on GPUs?

- Grouped convolution has already been used in Evolved Transformer [1] and Primer [2]. The paper should cite the related work and compare with the related work.

- This work demonstrates the performance on a unconventional Graphcore's Intelligence Processing Unit (IPU), which can benefit ops of lower computational intensity (like the Grouped FFN and convolution). The reviewer is not sure how useful the customized model is when the hardware becomes GPUs or TPUs.

[1] The Evolved Transformer, https://arxiv.org/abs/1901.11117

[2] Primer: Searching for Efficient Transformers for Language Modeling, https://arxiv.org/abs/2109.08668


**Summary Of The Paper:**

The paper proposed a set of modifications to the structure of the Transformer layer that improves FLOPS utilization. The proposed GroupBERT relies on grouped matrix multiplications and convolutions, being more efficient and superior in task performance and computation.

The overall network architecture contains four modulesL a multi-head attention, a grouped convolution module, and two grouped feed-forward modules (GFFN). Even being twice as many modules, the total FLOP increases of about 60% when sparse grouped operations are used.



**Summary Of The Review:**

Overall, the reviewer thinks the contributions are small. Grouped FFN and grouped convolution are not new. The paper lacks a comparison on MLM perplexity and downstream tasks with stronger baselines, such as evolved transformer and Primer where grouped ops are used. With respect to evaluation, only SQuAD is evaluated, which shows limited impact on generation tasks. The reviewer would suggest to evaluate on a wider set of down stream tasks such as GLUE and superGlue.

A few more detailed comments:
- Try to add more stronger baselines of efficient transformers (Reformer, Linformer, Evolved Transformer, Synthesizer, etc.).
- Add more tasks (GLUE or superGLUE)
- Target general accelerators such as GPUs. Then analyze training convergence time on a general processor.

---

> ### Author Response · Authors · 2021-11-19
> **Reply to 8smw**
>
> We thank the reviewer for providing insightful feedback about our work.
>
> To address your comments individually:
>
> - “This work demonstrates the performance on a unconventional Graphcore's Intelligence Processing Unit (IPU), which can benefit ops of lower computational intensity (like the Grouped FFN and convolution). The reviewer is not sure how useful the customized model is when the hardware becomes GPUs or TPUs.”
> “- Target general accelerators such as GPUs. Then analyze training convergence time on a general processor.”
>
> To complement model performance results on the IPU device, we present a comparison against FLOPs that shows a hardware agnostic point of view, for readers not in possession of an IPU.  More generally, suitability to GPU/TPU is not part of the ICLR call for papers and filtation on models according to this metric only exacerbates the effect of Hardware Lottery (Hooker [1]). With an increasing number of alternatives to these memory-bottlenecked architectures, this demand can be damaging to the pace of innovation in the field of machine learning.
>
>
> - “The ideals look trivial to the reviewer. Grouped FFN is not new and the structured sparsity formulation is not surprising. The hardware performance implication of the structured sparsity is not clear, as the paper is evaluated on a novel hardware accelerator. What are the performance implications when evaluating on GPUs?”
>
> We would like to point out that the grouping scheme proposed in this paper is novel and the authors are not aware on any work that used the same scheme. We apply the grouping to only one FFN matrix and compensate it with a small output projection (akin to post-attention projection) to combine the information that was processed individually by every group. To the best of our knowledge, such implementation has never been implemented by other works. More importantly, we show that it is a FLOP efficient way to apply grouped transformation, unlike previous implementations (e.g. SqueezeBERT).
>
> The implementation efficiency is dependent not only on a particular hardware type, but also software stack. While we have not evaluated our model on GPUs/TPUs, the implementation which will be available after the review period is based on out-of-the box TensorFlow functions and does not require any additional custom ops. Baseline BERT implementation is publicly available for IPU as well.
>
> - “Grouped convolution has already been used in Evolved Transformer [1] and Primer [2]. The paper should cite the related work and compare with the related work.”
>
> Both works used depthwise convolutions, which are not group convolutions. Group convolution generalises the idea of depthwise convolution, treating it as a special case with group_size=1. We would like to point out that Primer was released on arxiv on 17.09.2021, which is too close to the ICLR submission deadline to make an extensive comparison. Moreover, these models have not been proposed for bidirectional training.
>
> - “- Try to add more stronger baselines of efficient transformers (Reformer, Linformer, Evolved Transformer, Synthesizer, etc.).”
>
> Models that are listed above, with the exception of Evolved Transformer, target the O(n^2) complexity of the attention module, which was unchanged in GroupBERT. Therefore, these important contributions are fully compatible with GroupBERT and are orthogonal to our angle of exploration. Evolved transformer does not use grouped transformations, therefore it was out of scope of this investigation.
>
> - “-Add more tasks (GLUE or superGLUE)”
>
> We agree with the reviewer’s comments, and we will provide a range of GLUE results before the end of the discussion period.

---

> > ### Author Response · Authors · 2021-11-22
> > **GLUE results**
> >
> > We have conducted a range of experiments with BERT and GroupBERT families of models on GLUE datasets, which indicated there is a technical issue in our implementation of glue downstream tasks. We are working to fix this and will update the camera ready version with these scores.

---

### Official Review · Reviewer_s6Mm · 2021-11-08

**Correctness:** 3
**Technical Novelty And Significance:** 2
**Empirical Novelty And Significance:** 2
**Recommendation:** 5
**Confidence:** 3

**Details Of Ethics Concerns:**

No ethic concerns

**Main Review:**

The proposed architectural changes are simple but effective. The resulting model can be trained faster than the transformer baseline with the same MLM validation loss objective. However, the novelty of this paper is limited because many components such as GLU and convolutional layers have been proposed in the literature of efficient transformers. My major concerns are listed below and will consider raising the score if the major concerns can be addressed.

1. Comparison with the transformer baseline: As shown in Figure 1, it seems that the proposed GroupBERT is twice deeper than the transformer baseline. Since the transformer model scales very well with depth, it is also possible to reduce the width of the transformer and increase the depth to improve efficiency.
2. Empirical evaluation: There are no results on GLUE datasets. Moreover, since SQuAD v2 is more challenging than SQuAD v1.1, it is important to report the results on SQuAD v2 as well.
3. Comparison with prior arts: Figure 4a suggests that both prior arts achieve a worse computational cost and performance tradeoff compared to the transformer baseline. However, both papers claimed huge improvement over the BERT baseline. Why there is a gap between the results reported in this paper and the original paper. In addition, ConvBERT is a pre-work that should be compared to.
4. Number of groups: the authors claimed that group size 4 is optimal. Is it optimal for models with different sizes?


**Summary Of The Paper:**

In this paper, the authors propose applying both grouped feed-forward layers and grouped convolutional layers to improve the computational efficiency of the transformer model.

**Summary Of The Review:**

In this paper, an efficient transformer structure is proposed to improve the efficiency of the transformer model. However, some empirical evaluations are missing and comparisons with prior art are not convincing.

---

> ### Author Response · Authors · 2021-11-19
> **Reply to s6Mm**
>
> We thank the reviewer for providing insightful feedback about our work.
>
> To address your comments individually:
>
> - “Comparison with the transformer baseline: As shown in Figure 1, it seems that the proposed GroupBERT is twice deeper than the transformer baseline. Since the transformer model scales very well with depth, it is also possible to reduce the width of the transformer and increase the depth to improve efficiency.”
>
> For both GroupBERT and BERT families the number of layers and width is scaled in a similar way. The scaling behaviour of the baseline BERT family is defined in Turc et al. [1] where the relationship between width and depth was thoroughly studied, producing an optimal size scaling for BERT. Besides, if comparing the number of modules in a transformer, GroupBERT efficiency is clearly visible: if one compares BERT Large and GroupBERT Base, they have the same number of residual modules (48 modules), yet GroupBERT achieves the same MLM performance while using smaller width (and resulting >50% reduction of FLOPs and time-to-train).
>
> - “Empirical evaluation: There are no results on GLUE datasets. Moreover, since SQuAD v2 is more challenging than SQuAD v1.1, it is important to report the results on SQuAD v2 as well.”
>
> We agree with the reviewer’s comments and we will provide a range of GLUE results before the end of the discussion period. In the meantime, we were able to collect SQuAD 2.0 values for GroupBERT Base and compare them with the baseline:
>
>
> | Model | SQuAD 2.0 Exact Match | SQuAD 2.0 F1  | FLOPs |
> | ----------- | ----------- | ----------- | ----------- |
> | BERT Base      | 72.7 | 75.7 | 6.1e19 |
> | GroupBERT Base   | 77.3 | 79.1 |8.8e19 |
> | BERT Large   | 78.7 | 81.9 | 1.9e20 |
>
> These results demonstrate that GroupBERT maintains its Pareto efficiency with the more challenging SQuAD 2.0 task. Exact Match for GroupBERT base almost match BERT Large, while saving 50% of time and computation.
>
> - “Comparison with prior arts: Figure 4a suggests that both prior arts achieve a worse computational cost and performance tradeoff compared to the transformer baseline. However, both papers claimed huge improvement over the BERT baseline. Why there is a gap between the results reported in this paper and the original paper. In addition, ConvBERT is a pre-work that should be compared to.”
>
> In the MLM comparison on Figure 4a we show the pretraining performance for DeLight and SqueezeBERT, as well as both BERT and GroupBERT. DeLight language modelling results were presented while training on a much smaller dataset then the one used for BERT: Wikitext-103 with 103M tokens vs BERT Wikipedia+Bookcorpus 3300M tokens. SqueezeBERT reported a degradation in their task performance when pre-trained without distillation. However, it was unclear whether it was an efficient trade-off compared to smaller versions of BERT. In this work we clearly show that previous approaches do not match the dense baseline, while GroupBERT achieves better performance.
>
> - “Number of groups: the authors claimed that group size 4 is optimal. Is it optimal for models with different sizes?”
>
> We have tested this hyperparameter on a range of different model sizes and concluded that maintaining the number of groups to be equal to the expansion ratio of the FFN is the optimal choice for all model sizes.
>
> [1] Turc et al., Well-Read Students Learn Better: On the Importance of Pre-training Compact Models, 2019.

---

> > ### Author Response · Authors · 2021-11-22
> > **GLUE results**
> >
> > We have conducted a range of experiments with BERT and GroupBERT families of models on GLUE datasets, which indicated there is a technical issue in our implementation of glue downstream tasks. We are working to fix this and will update the camera ready version with these scores.

---

> > > ### Comment · Reviewer_s6Mm · 2021-12-01
> > > **Thanks for your response**
> > >
> > > Thank you for providing more experimental results. The new results on SQuADv2 help to verify the effectiveness of the proposed model. However, I am still not fully convinced that the proposed architecture is superior to the state-of-the-art efficient transformer models, especially since these models also utilize similar grouped linear layers. I will keep my original score.

---

### Decision · Program_Chairs · 2022-01-20

**Decision:**

Reject

**Comment:**

The authors propose modifications to the Transformer architecture in BERT by using grouped FFN and an additional convolution module.
The paper doesn't have all the results and comparison that should be done for a model that has seen similar architecture modification in the previous papers. While it is not necessary to show improvements on multiple hardware systems, it is important to see comparisons to more, stronger baselines and metrics on the full downstream GLUE eval rather than just Squad to establish improvements.
A reject.